# No effect of repetitive tDCS on daily smoking behaviour in light smokers: A placebo controlled EMA study

Ilse Verveer *, Danielle Remmerswaal, Joran Jongerling, Frederik M. van der Veen, Ingmar H. A. Franken

Department of Psychology, Education and Child Studies, Erasmus School of Social and Behavioural Sciences, Erasmus University, Rotterdam, The Netherlands

* verveer@essb.eur.nl

## Abstract

### Introduction

The effectiveness of repetitive transcranial Direct Current Stimulation (tDCS) on reducing smoking behaviour has been studied with mixed results. Smoking behaviour is influenced by affect and context, therefore we choose to use mobile ecological momentary assessments (EMA) to measure changes in smoking behaviour after tDCS.

### Methods

In a randomized, placebo-controlled, between subject study, we applied tDCS bilaterally with the anodal electrode targeting the right DLPFC (https://clinicaltrials.gov/ct2/show/NCT03027687). Smokers were allocated to six sessions of either active tDCS ($n = 35$) or sham tDCS ($n = 36$) and received two sessions on three different days in one week. They were asked to keep track of their daily cigarette consumption, craving and affect in an application on their mobile phones for three months starting one week before the first tDCS session.

### Results

Number of smoked cigarettes a day progressively decreased up to one week after the last tDCS session in both conditions. Active treatment had no additional effect on cigarette consumption, craving and affect.

### Conclusions

In this exploratory study, repetitive bilateral tDCS over the DLPFC had no effect on daily smoking behaviour. Future research needs to investigate how motivation to quit smoking and the number of tDCS sessions affect the efficacy of repetitive tDCS.

**Data Availability Statement:** All relevant data are within the manuscript and its Supporting Information files.

**Funding:** The author(s) received no specific funding for this work.

**Competing interests:** The authors have declared that no competing interests exist.

## Introduction

Smoking is associated with serious health risks and causes approximately 8 million deaths worldwide each year [1]. Although the health risks of smoking are generally well known, 1.1 billion people of the global population are still smokers [1]. The maintenance of tobacco addiction may be explained by an interplay of increased reward processing for smoking cues and decreased self-control over addictive behaviours [2, 3]. One brain area that plays a crucial role in this interaction is the dorsolateral prefrontal cortex (DLPFC) by its involvement in top down-control over reward processing [4]. Non-invasive neurostimulation (NIBS) is designed to directly modulate brain activity in specific brain areas. It is therefore suggested that NIBS over the DLPFC could enhance cognitive control of executive functioning, hereby reducing craving and substance use [5].

Transcranical Direct Current Stimulation (tDCS) is a well-tolerable NIBS that has no known serious adverse effects [6]. tDCS modulates membrane potentials in the brain by means of small electrical currents [7]. The electrical current flow from the anodal electrode to the cathodal electrode produces an electrical field that modulates the excitability of underlying brain areas [8]. This modulation of excitation levels is able to induce changes in behaviour, mood and cognition [9]. Also, cognitive control processes related to substance use disorder can be affected by tDCS [10]. Importantly, several studies on addictive behaviours have shown that tDCS could reduce craving. This effect has been found for a variety of substances, such as tobacco [11, 12], marijuana [13], cocaine [14], heroin [15], and alcohol [16]. For tobacco addiction specifically it was found that tDCS could not only reduce craving, but also cigarette consumption [17–21].

In a double-blind, sham-controlled, crossover study, Fecteau and colleagues [17] found that five tDCS sessions on consecutive days could decrease cigarette consumption for up to 4 days in participants who wanted to quit smoking. In another study where smokers were not planning to quit smoking in the next three months, results showed that cigarette consumption temporarily decreased after one session of tDCS [18]. Recently, it was found that five sessions of tDCS could decrease cigarette consumption for up to 4 weeks [19]. Here, motivation to quit modulated the effect of active tDCS on cigarette consumption. The results of these studies suggest that a variety of tDCS protocols could cause a decrease in cigarette consumption and craving. In addition, it was found that multiple sessions of tDCS may even provide a promising substitute to bupropion treatment in tobacco addiction [20]. Findings from a recent meta-analysis indicate that anodal tDCS over the right DLPFC with cathodal tDCS over the DLPFC had the most positive effects on smoking behaviour [21].

However, the exact parameters of the effectiveness of this specific tDCS protocol remain unknown. For example, it is unclear how many sessions are needed for tDCS to be effective in tobacco addiction and for how long the effects last beyond one-month follow-up. Recently, it was shown that 20 sessions of anodal tDCS over the left DLPFC may reduce cigarette consumption beyond one-month follow-up [20]. The current study will explore whether the protocol with anodal tDCS over the right DLPFC can also have extended effects on smoking behaviour with fewer sessions. *Ad libitum* smokers were included to pilot whether tDCS affects the natural course of smoking behaviour, without smokers being motivated to quit.

For the current study we choose to measure smoking behaviour by means of Ecological Momentary Assessment (EMA). Effects of tDCS on addictive behaviour have often been measured with retrospective self-reports in the lab. Since craving and substance use are both episodic phenomena that are associated with affect and context [22–24], measuring these variables in daily life may lead to more reliable answers. Furthermore, retrospective measurements may be influenced by recall biases [25, 26]. EMA therefore establishes more ecologically

valid results as compared to retrospective self-reports by collecting data in real-time repeatedly.

In sum, the aim of the current study is to explore the duration of the effect of repetitive tDCS on cigarette consumption by means of EMA in a sample of *ad libitum* smokers. Following the design of Falcone and colleagues [18], participants were included if they had no plans to actively try to quit smoking in the next three months. In line with previous studies, we expected that active tDCS can reduce the number of daily smoked cigarettes and we hypothesized that this decrease can last for up to 3 months after the last session. We also expected reduced craving after active tDCS [27] during and after the intervention week, and at three months follow-up.

## Materials and methods

### Participants

Seventy-three participants signed the informed consent form and completed the first tDCS session. Inclusion criteria were: 1) Between the age of 18 and 65 years; 2) Currently smoking 10 cigarettes or more a day; 3) The ability to speak, read, and write in Dutch. Exclusion criteria were: 1) Current substance use disorder of a substance other than nicotine or caffeine; 2) History of neurological or psychiatric disorders; 3) Any contraindication for electrical brain stimulation procedures (i.e. electronic implants or metal implants); 4) Pregnancy or breastfeeding; 5) Intentions to actively try to quit smoking in the next three months. Participants were recruited via advertisement at Erasmus University Rotterdam from October 2016 until March 2018 and received either course credit or a financial compensation of 20 euro. Two participants dropped out during the intervention week, because of personal circumstances (n = 1) and because of schedule issues (n = 1). Also due to schedule issues, three participants received tDCS on only two instead of three days. Leaving these participants out of analyses had no effect on the results, therefore they were included in the final analyses. Nine participants could not be reached after three months and were therefore lost to follow-up, leaving a total of 62 participants for follow-up analyses (Fig 1).

The study was approved by the Medical Ethics Committee of the Erasmus Medical Center, Rotterdam, the Netherlands. All procedures were carried out after participants were fully informed and had signed a written informed consent form. This report is part of the pre-registered study with identifier NCT03027687 at ClinicalTrials.gov. The complete study protocol can be found at http://dx.doi.org/10.17504/protocols.io.bcgdits6.

### Experimental design

The current study had a double-blind, randomized, sham-controlled design in which subjects received a total of six tDCS sessions (active or sham) on three days in one week with at least one day in between (Fig 2). Participants were first randomly assigned to either sham or active tDCS. Then, before the tDCS sessions and at three months follow-up, participants completed the Fagerström Test of Nicotine Dependence (FTND; [28]). Breath carbon monoxide concentrations were also measured using a Micro+ Smokerlyzer (Bedfont Scientific Ltd., Rochester, UK) to objectively define smoking. In addition, participants completed two behavioural tasks in the Erasmus Behavioural Lab (EBL) before the first session, one day after the last tDCS session and at three months follow-up. With these tasks we measured changes in cognitive control and feedback processing by means of EEG. The results of the tasks will be discussed elsewhere, in order to remain focus on the scope of this paper (e.g. changes in smoking behaviour after tDCS).

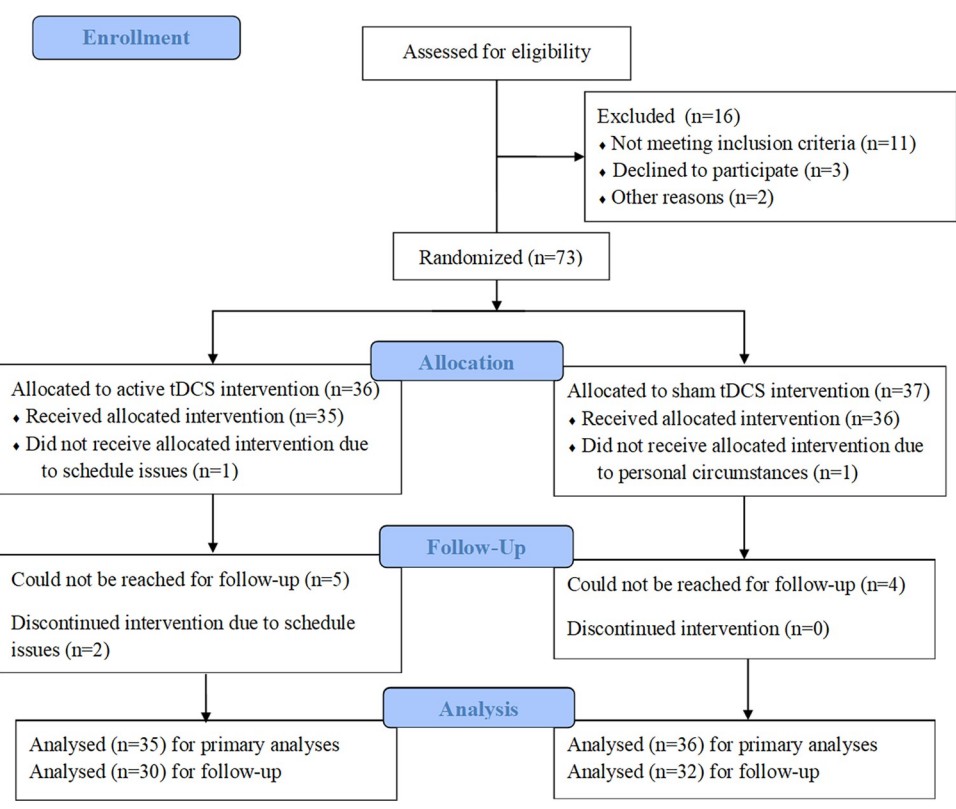

**Fig 1. Flow diagram.**

To measure changes in smoking behaviour, participants were asked to keep track of their cigarette consumption, craving and affect in an application on their mobile phones (EMA). Questions in the application were presented at four random times a day for three weeks in total (random assessments; RA's), starting the week before the first tDCS session (T1). After three months, participants were asked to fill out the same four-time daily random assessments for one more week (T2). In addition, participants were asked to start a session every time they smoked a cigarette (user-initiated smoking assessment; SA) for three months in total. An 'end

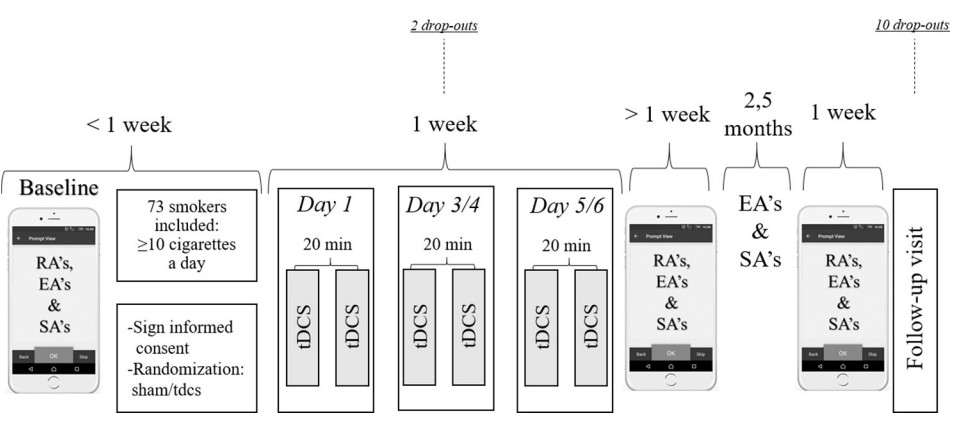

**Fig 2. Experimental procedure.**

of the day' assessment (EA) was also implemented for three months, which asked participants to fill out the total number of smoked cigarettes of that day.

Participants are asked to fill out the following EMA assessments: Random assessments (RA's), end of the day assessments (EA's), user-initiated smoking assessment (SA's). TDCS sessions last 13 minutes each with 20 minutes in between.

## Transcranial Direct Current Stimulation (tDCS)

Participants in the active tDCS group received tDCS by an electric DC-plus stimulator (Neuro-Conn, Ilmenau, Germany) via a pair of carbonated silicone electrodes with a thick layer of high-conductive EEG gel underneath them (35 cm$^2$). During each session, tDCS was applied two times for 13 minutes with a 20-minute break in between, and a current intensity of 2.0 mA with a 30 sec ramp at the beginning and end of the session [14]. The anodal electrode was placed over F4 area (10–20 international system) to stimulate the right DLPFC, the cathodal electrode was placed over the F3 (left DLPFC). Beneficial effects were found on smoking behaviour with this 'right anodal/left cathodal' positioning [17].

The control group received sham tDCS by the DC-plus stimulator. For sham, the electrodes were positioned at the same locations as active tDCS, but in this case the stimulator was gradually turned off after 30 s. Since the itching sensation of tDCS is often only experienced initially during stimulation, subjects remained blinded of the stimulation condition they received [e.g. 8, 29]. The experimenter was also blinded from the tDCS condition. That is, the codes that can automatically activate sham or active tDCS, were randomly assigned to participant numbers by an independent researcher. Then, the experimenter assigned the participant numbers.

## Ecological momentary assessment

**Procedure.** The LifeData platform (**www.lifedatacorp.com**) was used to develop the application for this study and to securely collect data. Participants were instructed by email to download the LifeData application on their smartphone one week before the first tDCS session. The start-up session of the application provided general information about how to use the app. After participants had finished the start-up session, they received random prompts four times a day between 10 am and 10 pm for 21 consecutive days to complete a RA. After three months, the application automatically started prompting participants again for four times a day for seven consecutive days. All RA's that were not completed within 90 minutes after the notification disappeared and were marked as missed.

In addition, participants were asked to initiate an assessment whenever they started smoking a cigarette (SA). The application further alerted participants at the end of the day (22 pm) to fill out the total number of cigarettes they had smoked during the day (end-of-day assessment; EA). EA's were prompted for 90 consecutive days, starting from the day the application was downloaded.

**Measures.** *Cigarette consumption.* During RA's, participants were asked how many cigarettes they had smoked since the last assessment and how many minutes had passed since they had smoked their last cigarette. Participants were also instructed to start a SA whenever they smoked a cigarette. During EA's participants filled out the total amount of cigarettes they had smoked during the day.

*Craving.* During RA's, participants were asked to indicate the urge to smoke a cigarette at that moment on a Likert scale ranging from 0 (*no urge*) to 100 (*very strong urge*).

*Mood.* General mood was measured during RA's by a prompt stating: "What is your general mood at the moment?" with response possibilities ranging from *very negative* (0) to *very positive* (5). In addition, participants were asked to evaluate the following specific affects for

themselves on a 5-point Likert scale: Happiness, enthusiasm, relaxedness, irritability, sadness, stress, and how bored they felt.

## Data analyses

In order to fit the nested data structure of Time within individuals (Level 1), and Group (tDCS vs. Sham) at Level 2, the primary analysis was conducted using multilevel regression modeling, also known as hierarchical linear modeling (see [30] for further details), in HLM 7.01. By using the maximum likelihood estimation method in multilevel modelling of the EMA data, all data points of individuals with missing data could be analyzed [31]. Missing data is almost inevitable in EMA studies, since most participants miss at least some prompts due to daily activities.

For the analyses, first a baseline model was fitted to every outcome variable (cigarette consumption and craving), including random intercepts across participants. With this model, it was assessed whether multilevel analysis was required. By significant variance at Level 2, the other models were fitted. It was confirmed that multilevel analyses could be applied in the current study, because fitting the baseline models to the data showed there was a significant amount of variance of the regression coefficients on the subject level (Level 2). The second model included the Level 1 predictor Time as fixed effect and was then extended by adding random slopes for Time. The final model included cross-level interactions between Time at Level 1 and the predictor variable Group (Active or Sham tDCS) at Level 2. The assumptions of normality and linearity were assessed by inspecting the residuals of each best fitted model. Unless otherwise reported, the assumptions were met. Further analyses examined smoking behaviour as a function of craving, and positive and negative affect on the momentary level (Level 1).

Since age and craving significantly differed for the sham and active tDCS group, multilevel analyses with cigarette consumption as outcome variable were also carried out with age and craving as covariates. Both covariates did not influence the results of tDCS on cigarette consumption. Explorative analyses were performed with the following covariates: Gender, overall FTND scores at baseline, and number of years the participant had been smoking. These variables had no influence on the effect of tDCS on cigarette consumption and craving. Finally, Spearman correlation coefficients were calculated for all three carbon monoxide scores on the one hand and mean number of cigarettes indicated by EA's in the week before each carbon monoxide concentration was measured on the other hand.

## Results

### Descriptive statistics

The final sample consisted of 71 participants (36 females, 35 males) between the age of 19 and 53 years (M = 22.3, SD = 4.7) who smoked an average of 11.3 cigarettes a day (SD = 4.2) and had a mean FTND score of 3.4 (SD = 1.9). Of these 71 participants, 35 received active tDCS and 36 received sham treatment. Because of the double-blind method the groups were not matched at baseline. As a result, the average age of the sham group (M = 23.4) was slightly higher compared to the active tDCS group (M = 21.1), $t(69) = 2.119$, $p = .038$. In addition, the active tDCS group experienced more craving at baseline ($p = .001$). For follow-up analyses, 62 participants were included ($n = 30$ Sham tDCS, $n = 32$ Active tDCS).

## Ecological momentary assessment: Compliance

RA's were initially prompted four times a day for 21 days, which means that the total number of possible prompts for 71 participants was 5964. The total number of completed RA's during the first 21 days was 2650. Therefore, the compliance rate for completed random assessments was 44.4%. Ninety days after the start-up session, RA's were prompted for one more week. During this follow-up week the compliance rate was 46.6%.

In addition, EA's were presented on each day for 90 days. During the first 21 days, participants completed a total of 810 out of 1491 EA's (54.3% compliance rate). During the follow-up week, 62 participants completed 242 out of 434 possible EA's, making the compliance rate 55.8%.

Additional exploratory analyses showed that 54 participants filled out at least one third of all EA's, and 38 participants filled out at least 50% of all EA's. Further analyses with these two groups showed no difference in results on the primary outcomes as compared to analyses with the entire sample. In addition, compliance on EA's did not correlate with the outcome measures for both groups.

## Primary outcome: Number of smoked cigarettes

The primary outcome measure was mean number of smoked cigarettes a day. Multilevel analysis with mean number of smoked cigarettes as dependent variable and time in days as predictor, showed that the mean number of smoked cigarettes slightly decreased over two weeks' time from the first tDCS intervention up to one week after the last tDCS session ($b = -.07$, $t(471) = -2.086$, $p = .038$). This decrease over time was observed for both active tDCS and sham tDCS (Fig 2) and did not correlate with EMA compliance. Importantly, no differences were found between the groups in the amount of change over time on number of smoked cigarettes ($p = .745$). Also at follow-up, the sham tDCS and active tDCS group did not differ in number of smoked cigarettes ($p = .859$).

## Correlational analyses: CO scores

Additional analyses showed that number of smoked cigarettes in the week before the first tDCS session correlated with the breath concentration of carbon monoxide (CO; in parts per million) on the day of the first tDCS session ($r = .416$, $p < .001$). CO scores on the day of the last tDCS session also correlated with the mean number of cigarettes smoked during the intervention week ($r = .303$, $p = .041$). Finally, it was found that mean number of smoked cigarettes in the week before the follow-up session correlated with CO scores at follow-up ($r = .486$, $p < .001$).

## Secondary outcomes: Craving and affect

Participants in the active tDCS group experienced significantly more craving ($p < .001$) the week before the first tDCS session ($M = 56.1$, $SD = 20.7$) as compared to the sham tDCS group ($M = 48.7$, $SD = 19.6$). Multilevel analysis with craving for cigarettes as dependent variable and time in days as predictor, showed a main effect of group in the two weeks after the first tDCS session ($b = 11.75$, $t(69) = 3.87$, $p = .003$), meaning that the baseline difference in craving was maintained throughout T1. There was no main effect of time ($p = .184$) and no interaction effect of time and groups ($p = .970$) on craving at T1 (Fig 3). No differences between groups were found for overall mood at T1 ($p = .599$). For T2 at 3 months follow-up, no main effect of time and condition, or interaction effect was found for craving and overall mood.

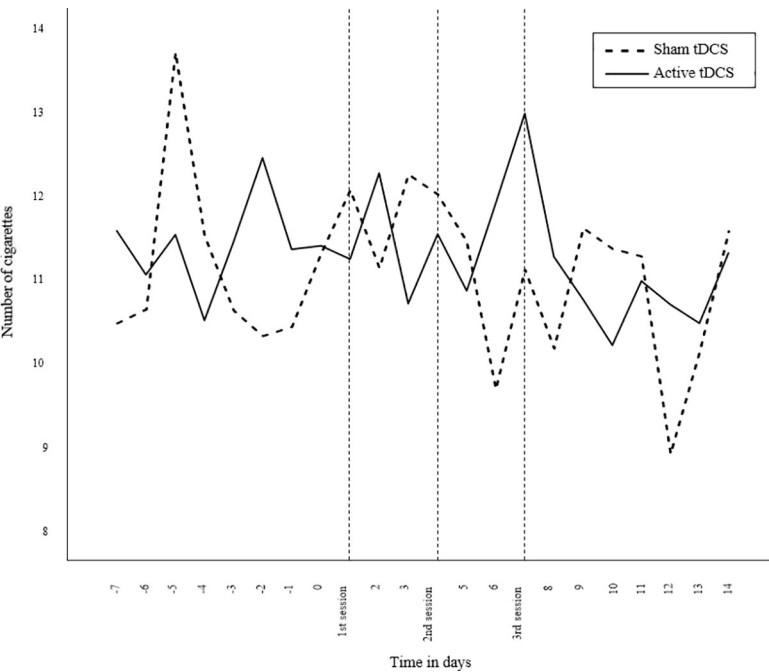

**Fig 3. Number of smoked cigarettes on each day starting from one week before the 1st tDCS session.**

Analyses at T1 showed that total number of smoked cigarettes was associated with craving on the same day ($b = .035$, $p < .001$). In addition, smoking behaviour was related to positive affect. Specifically, it was found that happiness was positively related to total number of smoked cigarettes on the same day ($b = .726$, $p < .001$).

## tDCS side effects

Side effects were recorded after each tDCS session. Participants were asked to indicate the amount of itching, burning, and tingling sensations on a 5-point Likert scale, ranging from none (1) to extreme (5) sensations. In addition, we asked participants about difficulties with concentrating and whether they experienced acute mood changes during tDCS. Questions about sleepiness, neckpain and pain in the head were also rated on a 5-point Likert scale. Results showed that overall, the active tDCS group experienced significantly more itching sensations ($F = 11.379$, $p = .001$) as compared to the sham group. No other differences between the groups were observed regarding the side-effects.

## Discussion

The current study was the first to use mobile Ecological Momentary Assessments (EMA) to investigate non-invasive neurostimulation as a tool to reduce smoking behaviour. The aim of this exploratory study was to test with EMA whether tDCS over the right DLPFC could modulate cigarette consumption and craving in *ad libitum* tobacco smokers. Results showed that over the course of the intervention week, the number of smoked cigarettes decreased during the application of sham or active tDCS. This finding is consistent with observations from a previous study with multiple tDCS sessions in tobacco smokers [17] and may result from the participants' awareness of their smoking behaviour.

Most importantly, however, no differences were found between the sham and active tDCS group in cigarette consumption and craving. This finding is not in line with a series of previous studies that have demonstrated that tDCS is effective in reducing cigarette smoking [17–20] and cigarette craving [11, 12]. A possible explanation for this unexpected finding is that in the current study craving and cigarette consumption were assessed by means of EMA. With this experience sampling method, we were able to measure cigarette consumption and craving in real time. The outcomes are therefore measured in a more ecologically valid manner as compared to retrospective self-reports. This is of importance, since smoking behaviour is associated with mood and context [23, 24]. Individuals may for instance smoke more during the weekend or on stressful days [31]. This pattern of change over time is clearly illustrated in Figs 3 and 4. Moreover, with the use of momentary assessments retrospective recall biases can be avoided.

Another explanation for the lack of effect of tDCS on craving levels and cigarette consumption can be found in the study's sample that consisted of mostly light smokers who, in addition, had no desire to quit smoking. It can be suggested that motivation of smokers to quit plays an important role in the efficacy of tDCS. The current study included *ad libitum* smokers (i.e., individuals who have no intention to quit smoking at the moment of the intervention), whereas a recent study that was published after we started our data collection showed that the effect of tDCS on smoking behaviour was modulated by motivation to quit [19]. It was also found that repetitive tDCS decreased cigarette consumption in participants who wanted to quit smoking [17]. These findings, in combination with the results of the present study, seem to suggest that tDCS is at least effective if there is a clear motivation to quit smoking. Future studies should explore the direct relationship between motivation to quit smoking and the efficacy of tDCS on smoking behaviour.

Finally, in contrast with previous studies that applied multiple sessions of tDCS, we applied 6 tDCS sessions on three different days in one week, instead of 5 or more session on at least five different days [e.g. 17, 20]. A reduction in cigarette consumption could nevertheless be expected on the days that tDCS was applied. That is, since Falcone and colleagues [18] found an immediate temporary effect of one tDCS session on smoking in *ad libitum* smokers. In this study, however, tDCS was applied online during cue exposure which may have influenced the effects [18].

Besides the important improvement of using real-time assessment in the natural environment of smokers, several critical remarks can be made and therefore caution should be taken when interpreting the findings. First, while participants were randomly allocated to either the active or sham condition, groups significantly differed on baseline craving levels. That is, participants in the active condition showed higher craving levels before the intervention compared to the control group which could have affected the results of this study.

A second limitation that should be mentioned is the relatively low compliance rate, ranging from 44% to 56%, on EMA assessments. A recent meta-analysis has shown that the average compliance rate in substance dependent samples is 69.8% [32]. Even though multilevel modelling in HLM 7.01 reliably corrects for random missing data, we performed additional analyses where participants with low compliance rates were excluded to investigate whether compliance rate might have influenced the outcomes. The results of these analyses indicated no change in outcome if compliance rates are higher. Reliability of the data is further supported by the finding that carbon monoxide concentrations correlated with number of smoked cigarettes as indicated in EMA end of the day assessments. Moreover, the EMA data showed that *ad libitum* smoking was related to craving and positive affect. Specifically, the number of smoked cigarettes increased with both craving and positive affect on the same day. These findings are in accordance with the results from an earlier EMA study with a higher compliance rate [22].

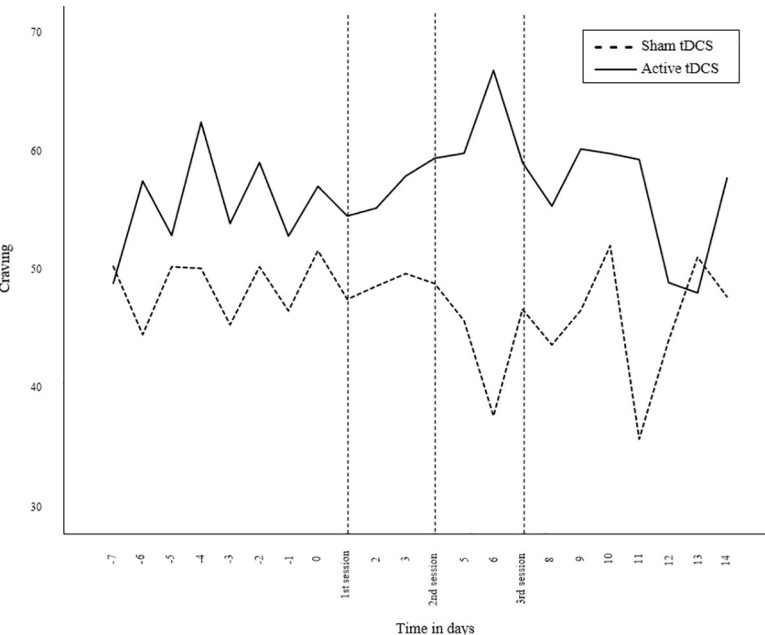

**Fig 4. Mean craving on each day starting from one week before the 1ˢᵗ tDCS session.**

Finally, participants in the active tDCS group indicated they experienced more itching sensations during neurostimulation than the sham tDCS group. This finding is in line with observations from previous studies where participants reported tingling and itching sensations after stimulation [e.g. 17]. However, blinding can still be reliable despite of differences in comfortability between the two conditions [33]. In addition, blinding with sham tDCS is considered reliable [29].

This was the first exploratory investigation using EMA to study the effects of tDCS on smoking behaviour. With the use of EMA, further insights were provided on the course of smoking behaviour over time. In sum, we did not find evidence that tDCS over the DLPFC decreases cigarette consumption and cigarette craving in light smokers that have no desire to quit at the moment of intervention. These findings raise intriguing questions regarding the nature and extent of the effects of tDCS on smoking behaviour. In a previous study it was found that motivation to quit smoking modulated the efficacy of tDCS on smoking behaviour [19], and therefore it may be necessary for smokers to actually quit smoking or at least be motivated to quit smoking at the moment of intervention. Future studies should explore this hypothesis by investigating the effects of repetitive tDCS in a larger sample of heavier smokers who are motivated to quit.

## Supporting information

**S1 Checklist. CONSORT 2010 checklist of information to include when reporting a randomised trial**∗.
(DOC)

**S1 File. Demographics.**
(SAV)

**S2 File. HLM Level 1 data at T1.**
(SAV)

**S3 File. HLM Level 1 data at T2.**
(SAV)

**S4 File. HLM Level 2 data for S2 File and S3 File.**
(SAV)

## Author Contributions

**Conceptualization:** Ilse Verveer, Danielle Remmerswaal, Frederik M. van der Veen, Ingmar H. A. Franken.

**Data curation:** Ilse Verveer.

**Formal analysis:** Ilse Verveer, Joran Jongerling.

**Investigation:** Ilse Verveer.

**Methodology:** Ilse Verveer, Danielle Remmerswaal, Frederik M. van der Veen, Ingmar H. A. Franken.

**Project administration:** Ilse Verveer, Ingmar H. A. Franken.

**Resources:** Ingmar H. A. Franken.

**Software:** Danielle Remmerswaal, Joran Jongerling.

**Supervision:** Danielle Remmerswaal, Frederik M. van der Veen, Ingmar H. A. Franken.

**Visualization:** Ilse Verveer.

**Writing – original draft:** Ilse Verveer.

**Writing – review & editing:** Ilse Verveer, Danielle Remmerswaal, Joran Jongerling, Frederik M. van der Veen, Ingmar H. A. Franken.

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
