## [Decision Letter · Decision Letter 0]

4 Sep 2019

PONE-D-19-19431

Effects of repetitive tDCS on ad libitum smoking behaviour: An EMA study

PLOS ONE

Dear Ms. Verveer,

Thank you for submitting your manuscript to PLOS ONE. After careful consideration, we have decided that your manuscript does not meet our criteria for publication and must therefore be rejected.

I am sorry that we cannot be more positive on this occasion, but hope that you appreciate the reasons for this decision.

Yours sincerely,

Berthold Langguth

Academic Editor

PLOS ONE

Reviewers' comments:

Reviewer's Responses to Questions

**Comments to the Author**

1. Is the manuscript technically sound, and do the data support the conclusions?

Reviewer #1: Yes

Reviewer #2: Yes

Reviewer #3: No

2. Has the statistical analysis been performed appropriately and rigorously? 

Reviewer #1: Yes

Reviewer #2: Yes

Reviewer #3: No

3. Have the authors made all data underlying the findings in their manuscript fully available?

Reviewer #1: No

Reviewer #2: Yes

Reviewer #3: No

4. Is the manuscript presented in an intelligible fashion and written in standard English?

Reviewer #1: Yes

Reviewer #2: Yes

Reviewer #3: Yes

5. Review Comments to the Author

Reviewer #1: The study presented to with a well rounded design, with a lot of sense of detail concerning means of control and potential critical aspects.

However, the following questions and recommendations arised:

Abstract:

- As lack of acute motivation to quit smoking is an important distinguishing aspect to related studies, it would be beneficial if it could be mentioned not only in the conclusions, but also in the introduction section of he abstract

Introduction:

- Please give further explanation on the underlying assumptions concerning the DLPFC and its role on reward processing and self-control as well as more detailed background information concerning hypothesized effects of tDCS on these mechanisms.

Methods:

- You state in line 119 that behavioural tasks where completed by the participants prior and after the tDCS sessions where conducted. While you report that the results and nature of these tasks are irrelevant for this paper, could you still please give more details?

- There are a few small punctuation errors (for example in line 167, 184, 220)

In line 168 probably “to be bored” is ment, rather then “boringness”.

- Was there a reason that lead to the decision to make intention to quit smoking an exclusion criteria? Please give more information on this topic.

Results:

- It could contribute to the information value, if the visual depiction and diagrams of the results would go along with a short inscription.

- Was there any noticeable relevance of the Fagerström test scores?

Discussion:

- Again, the lack of motivation could be, as you stated, an important modulating factor on the efficiancy of any quit smoking treatment approach. It should therfore be further elobarated on why the choice to exclude it was intentional and was of further benefit, compared to other study designs.

- In consideration of this, the assumptions you made in line 292- 297 seem risky.

Reviewer #2: This manuscript examines the effects of repetitive tDCS over DLPFC on smoking behavior using EMA(ecological momentary assessments). The idea was to measure smoking behavior in a "more ecologically valid manner as compared to retrospective self-reports". The main finding of this research is that both sham and tDCS groups show decreased cigarette consumption as well as craving, but no difference was found between the two groups.

General remark. As the authors mentioned, they hypothesized that tDCS over of the DLPFC would reduce cigarette smoking and craving as many others have reported, however, their results were negative. Although the authors argued that negative results can be valuable, but sole negative results are difficult to explain. There are many possible reasons for negative results.

1. The subjects were relatively light smokers.

2. It is hard to explain why right anodal/left cathodal DLPFC stimulation should have any effects.

As I understand, anodal stimulation increases brain activity whereas cathodal stimulation decrease brain activity. Unless left DLPFC and right DLPFC have totally opposite functions, why should we activate one side and inactivate the other side?

3. The baseline craving scores for the two groups were significantly different(p = 0.001). That makes it difficult to examine any effects of tDCS on craving.

4.Using EMA to measure cigarette consumption and craving is the novel part of this manuscript, however, the authors mentioned in the discussion that the controversial results might due to using EMA. As showed in the manuscript, EMA did not provide more information, especially how mood and context would influence the effects of tDCS. Low compliance rate is another problem.

5. The fact that cigarette consumption in Sham group decreased need to be explained.

6. There are no figure legends.

Reviewer #3: This is a study that seeks to determine the effect of tDCS on smoking behavior. Overall, the topic is important; however, a number of issues may limit the value of the article, I hope below comments improve the quality of the paper.

Title

According to the CONSORT statement, authors should indentify in the title that this is a controlled trial. Abbreviated words also should be removed from the title.

Abstract

1. Introduction is too long. Here the authors should use effectiveness instead of efficacy.

The sentence “To determine the duration of treatment effects, daily smoking behavior was studied for three months.” should be moved to method section.

2. The definition of all outcomes and statistical methods should be provided.

3. In contrast to introduction with unnecessary sentences, the results section is very brief with no effect sizes, confidence intervals and P values!

3. The conclusion should be according to presented results.

Introduction

1 Line 27, please provide the latest evidences

2. Lines 33-34, Please provide references for this sentence.

3. Line 44, marihuana or marijuana?

4. Line 46-52. There are still other related studies which should be cited such as:

https://www.sciencedirect.com/science/article/pii/S0924933819300793

5. Lines 56-63. Current study is also with limited sample size and follow up for three months is not long follow up, therefore it is not clear that what will be added this study to current knowledge.

6. I am not agreeing with authors that previous studies used retrospective measurements! with some degree of recall bias. Especially EMA is a subjective measurement with high probably of information bias.

Methods

1. No information provided about randomization type and method, concealment method, sample size calculation. Please consider CONSORT with caution.

2. The information about blinding and statistical methods is also insufficient. I am not believed that this study is double blind.

3. More details are required about assessments (i.e RA, SA and EA).

4. Line 121, why the results of behavioral tasks are beyond the scope of this paper?! Authors should provide all outcomes, declared in clinicaltrials.gov registration.

5. I am not sure that the method used for measurement of craving and mood are correct and based on DSM5.

6. Simple statistical methods such as independent t-test for changes over different intervals, repeated measured ANOVA and GEE are more appropriate especially for general readers. The methods for comparing baseline data should also be provided.

Results

1. It is unusual that with including participants who smoked more than 10 cigarettes per day, the mean daily cigarettes smoked become 11.29 with a SD equal to 4.2. Please provide the range of cigarettes smoked per day.

2. Line 194-198, with a true randomization there is no need for matching. Is this a randomized control trial?? The differences in age and craving of participants are attributed to inappropriate or lack of randomization although small differences between baseline variables may be present due to random error. Again please consider CONSORT and compare all studied variables at the baseline and different intervals.

3. The compliance rates for RA and EA are very low and this is a big limitation. Surprisingly authors did not compare these rates between intervention and control groups. Non respondents should be compare according to the other outcomes.

4. There are many fluctuations in number of smoked cigarettes in both groups (Fig 3). When one group show increased number of cigarettes other groups show decreased smoked cigarettes! This may be attributed to incorrect randomization, low compliance rate and information bias.

5. The results of statistical test should be provided in tables with all details.

6. Although the P-values are significant, the correlation coefficients between smoked cigarettes and CO are below 0.5 and contrary to the authors' claim it shows a low correlation.

Discussion

1. Lines 270-271, I am not agree that the data collected by EMA is more valid than previous studies. This study has many methodological errors.

2. Lines 282-283, Low compliance rate is an important issue. By excluding participants with low compliance, the sample size will be decrease (Although the total sample size was also low) and lack of significant difference between groups can be attributed to type two error.

3. Discussion should be revised after new and complete analysis of data and with addressing many limitations of study.

References

Should be updated.

Figures

Have low resolution and should be revised according to the new results.

6. PLOS authors have the option to publish the peer review history of their article (what does this mean?). If published, this will include your full peer review and any attached files.

Reviewer #1: No

Reviewer #2: No

Reviewer #3: No

- - - - -

---

## [Author Response · Author response to Decision Letter 0]

29 Oct 2019

The first two reviewers were quite positive and stated that we used an adequate methodology. The third reviewer was more critical but we believe that several responses of particularly this reviewer seem to indicate that there was some misconception about important aspects of the method and result section which we would like to address here. Considering all responses, the following seem crucial for the rejection of the current paper and were revised in the following manner:

1. Participants were light smokers:

This is correct, we were not only interested in heavy smokers, but all (including light) smokers. We have addressed this more clearly in the resubmitted paper.

2. Motivation to quit has not been taken into account: 

We did not take motivation into account since it only became apparent that motivation may influence the effects of tDCS after we had started the data collection. That is, the data collection started end of 2016 and the paper which indicates that motivation to quit may be an important modulation factor appeared in 2018. We therefore mention in our discussion that this might be a valid explanation for our null-findings. 

3. The sham and active tDCS group differed in craving at baseline:

This is an unfortunate result of our double-blind randomization procedure. To our surprise, the reviewer in question (3) mentions that “he/she is not believed that the study is double-blind”. In appeal to this response, we would like to refer to lines 141-143 where it’s stated that the tDCS system enabled the researcher to fill out a code which starts the procedure (either sham or active tDCS). The codes were assigned to participant numbers by an independent researcher. The procedure was therefore most certainly double-blind. We therefore believe that this baseline difference is the unfortunate result of chance. 

4. Compliance rate of EMA responses was low: 

This is correct and we agree. However, as mentioned by our review paper on this topic (Jones and colleagues; line 287 – 289) little is known about how to increase compliance rates. To correct for the low compliance rate, we performed exploratory analysis only including participants with a high compliance rate. This did not result in different outcomes. Other indicators of the reliability of our EMA data are also mentioned (line 290-298). 

Point-by-Point responses to the third reviewer:

Title and Abstract

The title and point 1-4 for the abstract have been adjusted in revisions. 

Introduction

Point 1-3: Indicated that small adjustments had to be made in certain sentences.

Point 4: The reviewer referred that the following paper should have been cited: https://www.sciencedirect.com/science/article/pii/S0924933819300793. However, this article was published in August 2019, whereas we submitted the manuscript in July. We have included the reference in the revised version of our manuscript.

Point 5: The reviewer mentioned that the study has a limited sample size and that 3 months is not a long follow-up. Still, the sample size is larger compared to previous studies with a between subject design that investigated the effects of tDCS on smoking behavior.

As far as we know, the largest sample for active tDCS in smokers was 19 participants (https://www.ncbi.nlm.nih.gov/pmc/articles/PMC5791546/pdf/fphar-09-00014.pdf) before we submitted the manuscript in Juli 2019, whereas we’ve included 35 participants for active stimulation. In addition, 3 months follow-up was the longest follow-up that had been performed in smokers so far. 

Point 6: The reviewer claimed that previous studies did not use retrospective self-reports. However, in comparison with ecological momentary assessments by which smoking behavior multiple is measured multiple times a day in different contexts, questionnaires at the end of the day or week (in the lab) are retrospective and could therefore affect recall. That is, since for example craving is a momentary phenomenon, and this sensation can be difficult to recall after a while. Participants also indicated that they became aware of how often they smoked by filling in the app, and that this was often more than they thought in retrospect.

Method:

Points 1 and 2: We would like to refer to line 104-106 and line 143-145, and the response above stating that our study design was randomized and double-blind. 

Point 3: Details about assessments are provided under “Procedure” and “Measures”.

Point 4: Behavioral tasks are beyond the scope of the paper because these also include EEG measures. It would make the article unnecessarily long and incomprehensive, and therefore we decided to publish the outcomes of the tasks in a separate article. 

Point 5: The reviewer mentions that simpler statistical methods could have been used. This is not the case since our data has a multilevel design with Time within participants at level 1 and Group at level 2, and should therefore be analyzed as such. We have addressed this more clearly in our revision (lines 176-182).

Results:

Point1: Participants were indeed lighter smokers. We have addressed this in the discussion (Line 300).

Point2 and 3: Randomization and compliance are addressed above.

Point 4: The reviewer mentions that many fluctuations in number of smoked cigarettes can be attributed to flaws in the research design. However, it’s more likely that smoking and craving (at least in light smokers) fluctuates over time according to mood and context. At the same time, this result therefore indicates the importance of using EMA measures.

All other points are related to above mentioned points or were small points that have been taken into account in the revision.

---

## [Decision Letter · Decision Letter 1]

31 Jan 2020

PONE-D-19-19431R1

No effect of repetitive tDCS on daily smoking behaviour as measured by EMA: A placebo-controlled study

PLOS ONE

Dear Ms. Verveer,

Thank you for submitting your manuscript to PLOS ONE. After careful consideration, we feel that it has merit but does not fully meet PLOS ONE’s publication criteria as it currently stands. Therefore, we invite you to submit a revised version of the manuscript that addresses the points raised during the review process.

As you can see, one reviewer still has Major concerns about your work. Please provide a revision, if you feel that you are able to address all concerns of the reviewer.

We would appreciate receiving your revised manuscript by Mar 16 2020 11:59PM. To enhance the reproducibility of your results, we recommend that if applicable you deposit your laboratory protocols in protocols.io, where a protocol can be assigned its own identifier (DOI) such that it can be cited independently in the future. For instructions see: http://journals.plos.org/plosone/s/submission-guidelines#loc-laboratory-protocols

We look forward to receiving your revised manuscript.

Kind regards,

Berthold Langguth

Academic Editor

PLOS ONE

and

Tifei Yuan

Academic Editor

PLOS ONE

2) Please include captions for your Supporting Information files at the end of your manuscript, and update any in-text citations to match accordingly. Please see our Supporting Information guidelines for more information: http://journals.plos.org/plosone/s/supporting-information

Reviewers' comments:

Reviewer's Responses to Questions

**Comments to the Author**

1. If the authors have adequately addressed your comments raised in a previous round of review and you feel that this manuscript is now acceptable for publication, you may indicate that here to bypass the “Comments to the Author” section, enter your conflict of interest statement in the “Confidential to Editor” section, and submit your "Accept" recommendation.

Reviewer #2: (No Response)

Reviewer #4: All comments have been addressed

2. Is the manuscript technically sound, and do the data support the conclusions?

Reviewer #2: Yes

Reviewer #4: Yes

3. Has the statistical analysis been performed appropriately and rigorously? 

Reviewer #2: I Don't Know

Reviewer #4: Yes

4. Have the authors made all data underlying the findings in their manuscript fully available?

Reviewer #2: Yes

Reviewer #4: Yes

5. Is the manuscript presented in an intelligible fashion and written in standard English?

Reviewer #2: Yes

Reviewer #4: Yes

6. Review Comments to the Author

Reviewer #2: Overall, this is not a well controlled study. The results are not sufficient to support the conclusion that rtDCS has no effects on daily smoking behavior. There are many possible reasons that the results are negative. For instance, the baseline of craving from two groups are significantly different; all subjects were light smokers so that there was the floor effect; main effects may be lost because of the low compliance rate, and so on.

Reviewer #4: In this study, the authors investigated the effectiveness of repetitive transcranial direct current stimulation to reduce smoking behaviour. The study was previously registered by clinicaltrials and has a sufficiently large sample (atDCS=35, stDCS=36) to answer the scientific questions raised. Participants received 6 prefrontal tDCS (anode F4, cathode F3, 2mA, 13 minutes) sessions on a total of three days. Outcome measures of smoking behaviour were collected using a mobile app with a frequency of responses (21 days, 4 times per day).

All in all, this is an interesting study that is well designed and sufficiently powered for a pilot study. There are still some minor issues:

- It makes sense to use age and craving as co-variables in the statistical model, but I don't understand why the gender was not included?

- Fig3/4: The SD for active and sham tDCS should be stored in the figure, e.g. less contrasting background.

P.S.: Another plus is the disclosure of the data in the supporting information (SPSS files).

7. PLOS authors have the option to publish the peer review history of their article (what does this mean?). If published, this will include your full peer review and any attached files.

Reviewer #2: No

Reviewer #4: Yes: Daniel Keeser

---

## [Author Response · Author response to Decision Letter 1]

13 Feb 2020

Response reviewer #2: Overall, this is not a well controlled study (1). The results are not sufficient to support the conclusion that rtDCS has no effects on daily smoking behavior (2). There are many possible reasons that the results are negative. For instance, the baseline of craving from two groups are significantly different (3); all subjects were light smokers so that there was the floor effect (4); main effects may be lost because of the low compliance rate (5), and so on:

1. The statement that the current study is not well-controlled seems very unfair and is unsubstantiated. The study had a double-blind, randomized, sham-controlled design (line 104-106) as explained in the following paragraph (line 139-145):

“The control group received sham tDCS by the DC-plus stimulator. For sham, the electrodes were positioned at the same locations as active tDCS, but in this case the stimulator was gradually turned off after 30 s. Since the itching sensation of tDCS is often only experienced initially during stimulation, subjects remained blinded of the stimulation condition they received [e.g. 8, 29]. The experimenter was also blinded from the tDCS condition. That is, the codes that can automatically activate sham or active tDCS, were randomly assigned to participant numbers by an independent researcher. Then, the experimenter randomly assigned the participant numbers.” 

This is the standard protocol for tDCS studies with a between subject design (see for example refs 14 and 19). We even preregistered the study; we have done everything in compliance to current standards. 

2. We have revised the title of the manuscript accordingly. We are careful not to explicitly state this conclusion and clearly describe in the discussion that the findings of the current study should be interpreted with caution because of several limitations (starting from line 300). The conclusion of the discussion points out that we refer only to the results provided in the current study, stating: 

 “In sum, in the current study we did not find evidence that tDCS over the DLPFC decreases cigarette consumption and cigarette craving in ad libitum smokers that 

 have no desire to quit at the moment of intervention.”

3. It is indeed the case that there are baseline differences in craving. This is however the result of randomization, as can happen in clinical trials. It does not assume a flaw in the design. We agree that it should be taken considered when interpreting the results. Therefore, we have taken craving at baseline into account as covariate and this did not change the results (line 191-193). In addition, the difference in craving remained over time (line 246-248). It would have been more problematic if we had found an effect of tDCS on craving, since that could have been attributed to a regression to the mean, but this was not the case.

4. We don’t agree that including light smokers is related to problems with the design of the study. We understand that our results in light smokers cannot by extrapolated to heavy smokers. In the discussion we mention that the smokers were light smokers and how this may have affected the results (line 288-299). We disagree that there was a floor effect, since cigarette consumption for both groups decreased over time (line 224-228). 

5. The EMA compliance rate was already discussed as a limitation, and further elaborated on in the discussion. We now slightly revised this to make it hopefully more clear (line 313 – 324):

“A second limitation that should be mentioned is the relatively low compliance rate of between 44.4% and 55.8% on EMA assessments. A recent meta-analysis has shown that the average compliance rate in substance dependent samples is 69.8% [31]. Even though multilevel modelling in HLM 7.01 reliably corrects for random missing data, we performed additional analyses where participants with low compliance rates were excluded to investigate whether compliance rate might have influenced the outcomes. The results of these analyses indicated no change in outcome if compliance rates are higher. Reliability of the data is further supported by the finding that carbon monoxide concentrations correlated with number of smoked cigarettes as indicated in EMA end of the day assessments. Moreover, the EMA data showed that ad libitum smoking was related to craving and positive affect. Specifically, the number of smoked cigarettes increased with both craving and positive affect on the same day. These findings are in accordance with the results from an earlier EMA study with a higher compliance rate [22].”

Response reviewer #4: In this study, the authors investigated the effectiveness of repetitive transcranial direct current stimulation to reduce smoking behaviour. The study was previously registered by clinicaltrials and has a sufficiently large sample (atDCS=35, stDCS=36) to answer the scientific questions raised. Participants received 6 prefrontal tDCS (anode F4, cathode F3, 2mA, 13 minutes) sessions on a total of three days. Outcome measures of smoking behaviour were collected using a mobile app with a frequency of responses (21 days, 4 times per day).

All in all, this is an interesting study that is well designed and sufficiently powered for a pilot study. There are still some minor issues:

- It makes sense to use age and craving as co-variables in the statistical model, but I don't understand why the gender was not included? (6)

- Fig3/4: The SD for active and sham tDCS should be stored in the figure, e.g. less contrasting background (7)

P.S.: Another plus is the disclosure of the data in the supporting information (SPSS files):

6. Gender was also included as covariate (line 193 – 195). In addition, the study included a similar amount of male and female participants and there were no differences of gender between groups (active and sham tDCS).

7. We have revised the figures accordingly. 

The main point we would like to make is that we don’t agree with reviewer #2 that there are flaws in the research design and/or methodology. The results may be disappointing - it did not show the hypothesized effects of tDCS - yet they are what they are. We hope that given the recent developments in improving science by facilitating the publication of null results when a study is well-designed (as reviewer # 4 also acknowledges) and the policy of PLOS ONE, the study will be published despite the fact that we were not able to find effects of tDCS in this population. We have addressed possible limitations clearly and critically in the discussion and therefore request that our resubmitted manuscript can be considered for publication in PLOS ONE.

---

## [Decision Letter · Decision Letter 2]

3 Apr 2020

PONE-D-19-19431R2

No effect of repetitive tDCS on daily smoking behaviour in light smokers: A placebo controlled EMA study

PLOS ONE

Dear Ms. Verveer,

Thank you for submitting your manuscript to PLOS ONE. After careful consideration, we feel that it has merit but does not fully meet PLOS ONE’s publication criteria as it currently stands. Therefore, we invite you to submit a revised version of the manuscript that addresses the points raised during the review process.

Please consider the Points raised by Reviewer 5 (statistical reviewer).

We would appreciate receiving your revised manuscript by May 18 2020 11:59PM. To enhance the reproducibility of your results, we recommend that if applicable you deposit your laboratory protocols in protocols.io, where a protocol can be assigned its own identifier (DOI) such that it can be cited independently in the future. For instructions see: http://journals.plos.org/plosone/s/submission-guidelines#loc-laboratory-protocols

We look forward to receiving your revised manuscript.

Kind regards,

Berthold Langguth

Academic Editor

PLOS ONE

Reviewers' comments:

Reviewer's Responses to Questions

**Comments to the Author**

1. If the authors have adequately addressed your comments raised in a previous round of review and you feel that this manuscript is now acceptable for publication, you may indicate that here to bypass the “Comments to the Author” section, enter your conflict of interest statement in the “Confidential to Editor” section, and submit your "Accept" recommendation.

Reviewer #2: All comments have been addressed

Reviewer #4: All comments have been addressed

Reviewer #5: (No Response)

2. Is the manuscript technically sound, and do the data support the conclusions?

Reviewer #2: Yes

Reviewer #4: Yes

Reviewer #5: Yes

3. Has the statistical analysis been performed appropriately and rigorously? 

Reviewer #2: Yes

Reviewer #4: Yes

Reviewer #5: Yes

4. Have the authors made all data underlying the findings in their manuscript fully available?

Reviewer #2: Yes

Reviewer #4: Yes

Reviewer #5: Yes

5. Is the manuscript presented in an intelligible fashion and written in standard English?

Reviewer #2: Yes

Reviewer #4: Yes

Reviewer #5: Yes

6. Review Comments to the Author

Reviewer #2: (No Response)

Reviewer #4: The authors have answered all my open questions. The revised manuscript has improved significantly. I have no further questions. However, I would like to emphasize the importance of non-result studies for the field. These are important and should be published. Daniel Keeser

Reviewer #5: The authors were able to address the previous comments raised satisfactorily. The study can be highlighted as a "pilot study" in the manuscript. If I am not mistaken, there is no sample size/power statement presented in the manuscript. On what basis was 73 participants included ? At this late stage, I am not asking authors to propose and justify this, but some comments in this regard (claiming this as a pilot study) would be helpful.

I have some additional minor comments. Although the statistical methods presented have been broadly classified as "multilevel methods" (see Page 10; Data Analyses section), more details are expected for the reader to understand better. Specificaly, are they fitting a linear (or non-linear) mixed effects model (pretty typical in repeated measures scenario)? The repeated measures in those models are usually factored in by introduction of random effects. It is unclear what they are doing; some more details would have assisted in a smoother reading.

7. PLOS authors have the option to publish the peer review history of their article (what does this mean?). If published, this will include your full peer review and any attached files.

Reviewer #2: No

Reviewer #4: Yes: Daniel Keeser

Reviewer #5: No

---

## [Author Response · Author response to Decision Letter 2]

1 May 2020

Response to reviewer #4: 

We would like to thank Daniel Keezer for his valuable comments, and we agree that the results of the current study are highly important in the field of neurostimulation in addiction and should be published.

Response to reviewer #5: 

Dear reviewer, thank you for raising these important questions.

1. Regarding the sample size: we have conducted a power analysis before the start of the study, which we indicated in our medical ethical submission. The following power statement was made:

“The number of participants was based on a previous study in which the effect of repetitive tDCS on cigarette consumption was examined in 12 participants using a within-subject design (Fecteau et al., 2014). Our sample size was estimated based on a moderate effect size of tDCS on cigarette consumption on the last day of their study, which was four days post treatment (d = -.58). Based on a two-tailed p-value of .05 and a power of .80, we need approximately 80 participants to find significant between-subject differences, if present.”

We agree that this sample estimation is subjective to bias, as their study design was different from ours and the outcomes were previously not measured by EMA. We therefore adapted the manuscript by mentioning the exploratory nature of the current study.

2. We have added information to the manuscript regarding multilevel modeling (lines 181-196) and referred to Hox (2010) for further information.

---

## [Decision Letter · Decision Letter 3]

6 May 2020

No effect of repetitive tDCS on daily smoking behaviour in light smokers: A placebo controlled EMA study

PONE-D-19-19431R3

Dear Dr. Verveer,

We are pleased to inform you that your manuscript has been judged scientifically suitable for publication and will be formally accepted for publication once it complies with all outstanding technical requirements.

With kind regards,

Berthold Langguth

Academic Editor

PLOS ONE

Additional Editor Comments (optional):

Reviewers' comments:

Reviewer's Responses to Questions

**Comments to the Author**

1. If the authors have adequately addressed your comments raised in a previous round of review and you feel that this manuscript is now acceptable for publication, you may indicate that here to bypass the “Comments to the Author” section, enter your conflict of interest statement in the “Confidential to Editor” section, and submit your "Accept" recommendation.

Reviewer #5: All comments have been addressed

2. Is the manuscript technically sound, and do the data support the conclusions?

Reviewer #5: Yes

3. Has the statistical analysis been performed appropriately and rigorously? 

Reviewer #5: Yes

4. Have the authors made all data underlying the findings in their manuscript fully available?

Reviewer #5: (No Response)

5. Is the manuscript presented in an intelligible fashion and written in standard English?

Reviewer #5: (No Response)

6. Review Comments to the Author

Reviewer #5: (No Response)

7. PLOS authors have the option to publish the peer review history of their article (what does this mean?). If published, this will include your full peer review and any attached files.

Reviewer #5: No

---

## [Editor Report · Acceptance letter]

11 May 2020

PONE-D-19-19431R3 

No effect of repetitive tDCS on daily smoking behaviour in light smokers: A placebo controlled EMA study 

Dear Dr. Verveer:

I am pleased to inform you that your manuscript has been deemed suitable for publication in PLOS ONE. Congratulations! Your manuscript is now with our production department. 

With kind regards,

on behalf of

Dr. Berthold Langguth 

Academic Editor

PLOS ONE